# Effects of Polar Steroids from the Starfish *Patiria (=Asterina) pectinifera* in Combination with X-Ray Radiation on Colony Formation and Apoptosis Induction of Human Colorectal Carcinoma Cells

**DOI:** 10.3390/molecules24173154

**Published:** 2019-08-29

**Authors:** Olesya S. Malyarenko, Timofey V. Malyarenko, Alla A. Kicha, Natalia V. Ivanchina, Svetlana P. Ermakova

**Affiliations:** 1G.B. Elyakov Pacific Institute of Bioorganic Chemistry, Far Eastern Branch of the Russian Academy of Sciences, 159 100-let Vladivostok Ave., 690022 Vladivostok, Russia; 2Department of Bioorganic chemistry and Biotechnology, School of Natural Sciences, Far Eastern Federal University, Sukhanova str. 8, 690000 Vladivostok, Russia

**Keywords:** starfish, *Patiria (=Asterina) pectinifera*, polar steroids, colorectal carcinoma cells, X-ray, colony formation, apoptosis, DNA fragmentation

## Abstract

Despite significant advances in the understanding, prevention, and treatment of cancer, the disease continues to affect millions of people worldwide. Chemoradiation therapy is a rational approach that has already proven beneficial for several malignancies. However, the existence of toxicity to normal tissue is a serious limitation of this treatment modality. The aim of the present study is to investigate the ability of polar steroids from starfish *Patiria (=Asterina) pectinifera* to enhance the efficacy of radiation therapy in colorectal carcinoma cells. The cytotoxic activity of polar steroids and X-ray radiation against DLD-1, HCT 116, and HT-29 cells was determined by an MTS assay. The effect of compounds, X-ray, and their combination on colony formation was studied using the soft agar method. The molecular mechanism of the radiosensitizing activity of asterosaponin P_1_ was elucidated by western blotting and the DNA comet assay. Polar steroids inhibited colony formation in the tested cells, and to a greater extent in HT-29 cells. Asterosaponin P_1_ enhanced the efficacy of radiation and, as a result, reduced the number and size of the colonies of colorectal cancer cells. The radiosensitizing activity of asterosaponin P_1_ was realized by apoptosis induction through the regulation of anti- and pro-apoptotic protein expression followed by caspase activation and DNA degradation.

## 1. Introduction

Colorectal carcinoma is the third most common cancer in men and the second in women. In 2018, there were in excess of 1.8 million new cases worldwide [1]. Radiation therapy using high-energy rays (such as X-ray) is the method most often used to destroy rectal cancer cells. In turn, surgery, chemotherapy, immunotherapy, and the combinatorial treatment by chemo- and radiation therapy, called chemoradiation therapy, are the most effective curative modalities for people with colorectal cancer [2]. However, the short- and long-term side effects of high doses of radiation exposure, as well as widely used chemotherapeutic drugs (5-Fluorouracil, Capecitabine, Oxaliplatin, and Trifluridine), are encouraging scientists to develop more effective schemes for cancer therapy [3]. The selective sensitization of cancer cells by natural, non-toxic compounds to low doses of ionizing radiation is a novel approach for improving cancer treatment.

Starfishes (phylum: Echinodermata, class: Asteroidea) are typical representatives of the marine benthic fauna. Their distribution extends from cold Arctic and Antarctic waters to the tropics, and they can be found in habitats ranging from reef communities to shallow water. This class of invertebrates is a rich source of various low-molecular weight metabolites: sterols, polyhydroxysteroids, steroidal glycosides, carotenoids, peptides, fatty acids, and anthraquinoid pigments, as well as sphingolipids and their derivatives [4]. The most common low-molecular weight metabolites of starfishes are polar steroids and sphingolipids.

The starfish *Patiria* (=*Asterina*) *pectinifera* is a widespread inhabitant of the Far Eastern seas; therefore, it has been an object of research for many years. Previously, steroidal hexaol, octaol [5], a polyhydroxysteroid glycoside named asterosaponin P_1_ [6,7], asterosaponins (pectiniosides A–F) [8,9,10], acanthacerebroside B [11], and gangliosides [12] were isolated and their structures established.

Low-molecular weight compounds of starfish *P.* (=*Asterina*) *pectinifera* are found to exhibit a variety of biological activities. For example, the (25*S*)-5α-cholestane-3β,4β,6α,7α,8,15β,16β,26-octaol and (25*S*)-5α-cholestane-3β,6α,7α,8,15α,16β,26-heptaol enhanced neuritis growth in a mouse neuroblastoma cell line possessed notable synergistic effects on nerve growth factor (1 ng/mL) or brain-derived neurotrophic factor (0.1 ng/mL), and exerted a neuroprotective effect against oxygen-glucose deprivation by increasing the number of surviving cells [13,14]. The asterosaponin P_1_ was shown to inhibit the activity of sulphatase from the mollusk *Turbo chrysostomus* [15]. Polar steroids from *P. pectinifera* were demonstrated to exert antiviral (HSV-1) and cytotoxic activity against liver hepatocellular carcinoma HepG2 cells in vitro [16].

The aim of the present work is to investigate the anticancer and radiosensitizing activities of polar steroids from *P. pectinifera* in a colorectal carcinoma cell model of colony formation and apoptosis induction.

## 2. Results and Discussion

### 2.1. Effect of Polar Steroids from P. pectinifera on Cancer Cell Viability 

In the first step of bioactivity investigations, the cytotoxicity of (25*S*)-5α-cholestane-3β,4β,6α,7α,8,15β,16β,26-octaol (**1**), (25*S*)-5α-cholestane-3β,6α,7α,8,15α,16β,26-heptaol (**2**), and asterosaponin P_1_ (**3**) (Figure 1) was determined by measuring the metabolic activity of human colorectal carcinoma cells DLD-1, HCT 116, and HT-29 using an MTS reagent. None of the tested compounds inhibited the viability of HCT 116, DLD-1, and HT-29 cells by 50% at concentrations up to 150 µM. Only compounds **1** and **3** (150 µM) slightly decreased cell viability (the percentage of inhibition was less than 30%) in 24 h of treatment (data not shown). Previously, it was reported that polyhydroxysteroids and steroidal glycosides from starfishes possess moderate cytotoxic activity against human cancer cell lines [17,18,19]. To detect the inhibitory effect on colony formation in cancer cells, compounds **1**–**3** were used in concentrations of 10–40 µM in further experiments.

### 2.2. The Effect of Polar Steroids from P. pectinifera on Colony Formation in Human Colorectal Carcinoma Cells Alone and in Combination with X-Ray Exposure

Colony formation is an ability of a single cancer cell to form colony via clonal expansion and is thought to be one of the chemotherapeutic hallmarks of carcinogenesis [20]. In the present study, the effect of polar steroids from *P. pectinifera* on colony formation in colorectal carcinoma cells was verified using a soft agar assay at the first time. The obtained results revealed that polar steroids **1**, **2**, and **3** (at concentration of 40 µM) decreased the number of colonies of DLD-1 cells by 15%, 14%, and 26%; HCT 116 cells by 9%, 5%, and 13%; and HT-29 cells by 18%, 17%, and 38%, respectively, compared to PBS-treated cells (control) (Figure 2A–C). The investigated compounds did not influence the size of colorectal carcinoma cell colonies (data not shown).

Among the investigated types of colorectal carcinoma cells, the most resistant cell line to the inhibitory effect of polar steroids **1**–**3** seemed to be HCT 116 cells, and the most sensitive cells were the HT-29 cells. Among polar steroids investigated in this study, steroidal monoside asterosaponin P_1_ had the highest inhibitory activity against colony formation in HT-29 cells. The obtained results are in accordance with previously published data, namely recent reports of polar steroids effectively suppressing colony formation in different types of human cancer cells with a greater extent of inhibition on human colorectal cancer cell lines [21,22].

Nowadays, strategies for decreasing the toxicity of radiotherapy by using effective, non-toxic radiosensitizers from natural sources are of great importance [23,24,25]. To determine the dose of X-ray radiation (ID_50_) that caused 50% inhibition of the number of cancer cell colonies, DLD-1, HCT116, or HT-29 cells were treated with X-ray at doses from 2 to 10 Gy, and after 3 h the irradiated cells were subjected to the soft agar assay as described in the section “Materials and Methods.” The sensitivity of tested colorectal cancer cells to X-ray increased in the panel of DLD-1 (ID_50_–10.6 Gy), HCT 116 (ID_50_–8.43 Gy), and HT-29 cells (ID_50_–4.65 Gy), resulting in a reduced number of colonies (Figure 2D). Moreover, the size of colonies under radiation exposure was reduced dramatically in the panel of HT-29 (ID_50_–7.0 Gy), HCT-116 (ID_50_–5.6 Gy), and DLD-1 cells (ID_50_–4.5 Gy) (Figure 2E).

We then investigated whether polar steroids from *P. pectinifera* could enhance the efficacy of low-dose X-ray irradiation in human colorectal carcinoma cell lines. To this end, DLD-1, HCT116, and HT-29 cells were treated with compounds **1**–**3** at concentrations of 1–4 µM and exposed to a non-toxic dose of 2 Gy, which alone did not exert an inhibitory effect on viability or colony formation in cancer cells.

Under X-ray exposure, the number of colonies of DLD-1, HCT 116, and HT-29 cells decreased by 10%, 15%, and 24%, respectively, compared to the control (Figure 3A–C). The size of the colonies of DLD-1, HCT 116, and HT-29 cells was reduced by 31%, 18%, and 7%, respectively, compared to the control (Figure 3).

Polyhydroxysteroids **1** and **2** (at concentrations of 1–4 µM) did not enhance the inhibitory effect of X-ray radiation on colony formation in the tested cell lines (Figure 3A–E). Asterosaponin P_1_ (4 μM) was found to sensitize DLD-1, HCT 116, and HT-29 cells to radiation by 13%, 8%, and 21%, respectively, compared to irradiated cells, resulting in a significant decrease in colony number (Figure 3A,C,E). Moreover, the colony sizes of DLD-1, HCT 116, and HT-29 cells were reduced by 7%, 12%, and 25%, respectively (Figure 3B,D,F). To the best of our knowledge, this is the first evidence that asterosaponin P_1_ from *P. pectinifera* possessed radiosensitizing activity in human colorectal carcinoma cells, resulting in significant inhibition of cancer cell colony formation.

Next, we elucidated the molecular mechanism of radiosensitizing effect of asterosaponin P_1_ (**3**) in HT-29 cells, the most sensitive cells of the tested cell lines.

### 2.3. Effect of Polar Steroids from P. pectinifera and X-Ray Radiation on Induction of Apoptosis in Human Colorectal Carcinoma Cells 

One of the molecular bases of radiation therapy is the damage to the genetic material (deoxyribonucleic acid, DNA) of cells, which suppresses proliferation and results in cell death (apoptosis) [26,27].

Apoptosis is a caspase-dependent process of individual cells death. It can be realized under extracellular (death receptor signaling pathway) or intracellular (mitochondrial signaling pathway) stimuli [28]. Apoptosis initiation through the mitochondrial signaling pathway generally occurs in response to cellular stresses such as DNA-damaging cytotoxic drugs or ionizing radiation. In this regard, proteins of the Bcl-2 family mediate mitochondrial damage. These proteins are usually classified according to a functional-domain basis for anti-apoptotic proteins of the Bcl-2 family (Bcl-XL, Bcl-w, Mcl-1, Bfl-1, Boo, Bnip-2, etc.), which inhibit the release of apoptogenic factors by the mitochondria, and pro-apoptotic (for example: Bax, Bak, Bad, Mtd, Diva, Bnip-Salpha, and Bnip-Sbeta), which cause their release [29]. The balance between the pro- and anti-apoptotic members of the Bcl-2 family regulates the release of pro-apoptotic substances from the mitochondria, such as cytochrome c. Cytochrome c in the cytoplasm is involved in the formation of the apoptosome, together with the protein APAF-1 and dATP, causing the activation of caspase 9. Activated caspase 9 subsequently activates caspase 3, leading to the generation of an apoptotic phenotype [30].

There is a large amount of published data on the ability of polar steroids from starfishes to induce apoptosis. Thus, the treatment of human glioblastoma U87MG cells by novaeguinoside II from the starfish *Culcita novaeguineae* was proven to result in the decreasing of the mitochondrial transmembrane potential and the up-regulation of cytochrome-c and caspase-3 expression, leading to the mitochondrial apoptosis induction [31]. Luzonicosides from the starfish *Echinaster luzonicus* inhibited the proliferation, colony formation, and migration of human melanoma cells. The molecular mechanism was associated with the cell cycle arrest, down-regulation of Bcl-2 protein, and the activation of effector caspase-3 [32]. Asterosaponin 1 was able to induce apoptosis in human lung cancer cells A549 by the regulation of GRP78 and GRP94 (molecular chaperones of endoplasmic reticulum (ER)) expression level and by increasing CHOP activity, caspase-4, and JNK kinase expression (important ER-associated proteins) [33]. Steroidal glycosides from the starfish *Anthenea aspera* exhibited pro-apoptotic activity via the regulation of the expression level of pro-survival protein Bcl-XL and pro-apoptotic proteins Bax and Bak, followed by the activation of the initiator and effector caspases in colon cancer cells [34].

Since polar steroids from starfishes induced apoptosis in different types of cancer cells, we checked the hypothesis as to whether asterosaponin P_1_ from *P. pectinifera* enhanced the radiation-induced apoptosis in colorectal carcinoma HT-29 cells by the activation of major players in mitochondrial apoptosis (Bax, Bcl-XL, caspase 9, and caspase 3).

As shown in Figure 4, X-ray radiation (2 Gy) slightly inhibited the expression of the pro-survival protein Bcl-XL, but did not influence the expression level of the pro-apoptotic protein Bax, initiator caspase 9, or effector caspase 3 in irradiated HT-29 cells. Combinatorial treatment of colorectal carcinoma cells with the asterosaponin P_1_ (1, 2 and 4 µM) and radiation (2 Gy) caused downregulation of the Bcl-XL protein and upregulation of the Bax protein in a dose-dependent manner that caused an activation of caspases 9 and 3, followed by proteolytic cleavage of caspase 3. The obtained data revealed that the asterosaponin P_1_ sensitized HT-29 cells to radiation by enhancing apoptosis induction. We suggested that the molecular mechanism of this action was associated with the downregulation of the pro-survival Bcl-XL protein and the upregulation of pro-apoptotic the Bax protein, leading to initiator and effector caspase activation.

The degradation of DNA molecules is a critical phase of apoptosis. In this study the radiosensitizing effect of asterosaponin P_1_ from *P. pectinifera* on the DNA degradation of HT-29 cells was studied by the DNA-comet assay.

This is simple and time-tested method that allows for the determination of DNA damage in different cells [35,36]. DNA ruptures were not observed in non-treated HT-29 cells (control). A low dose of X-ray irradiation (2 Gy) caused slight degradation of DNA, giving the observed objects the appearance of comets (indicated by white arrow). Accordingly, the number of comets corresponds to the number of strand breaks of DNA (Figure 4C). Quantitative analysis for the comet assay showed that the average tail moment (TM) for the HT-29 cells treated by X-ray alone was 8.613 ± 3.796 (n = 50, n—number of comets in each experimental group). Combinatorial treatment of HT-29 cells with asterosaponin P_1_ (4 µM) and X-ray (2 Gy) led to a significant increase in the number of DNA comets (indicated by white arrows) and the comet tail moment, indicating DNA degradation (TM was 29.516 ± 10.574, n = 50). These data provide evidence that the combination treatment by asterosaponin P_1_ and X-ray noticeably enhanced DNA damage in colorectal carcinoma compared to cells treated with X-ray alone, which confirmed the radiosensitizing effects of asterosaponin P_1_ (Figure 4C).

## 3. Materials and Methods

### 3.1. Reagents

Phosphate buffered saline (PBS), L-glutamine, penicillin-streptomycin solution (10,000 U/mL, 10 µg/mL), and trypsin were from Sigma-Aldrich (St. Louis, MO, USA). MTS reagent 3-(4,5-Dimethylthiazol-2-yl)-5-(3-carboxymethoxyphenyl)-2-(4-sulfophenyl)-2*H*-tetrazolium was purchased from Promega (Madison, WI, USA). Low- and normal-melt agarose were purchased from Helicon (Moscow, Russia). Basal Medium Eagle (BME), McCoy’s 5A Modified Medium (McCoy’s 5A), Roswell Park Memorial Institute medium (RPMI-1640), fetal bovine serum (FBS), agar, gentamicin, and Pageruler Plus Prestained Protein Ladder were purchased from ThermoFisher Scientific (Waltham, Massachusetts, USA).

Cell lysis buffer (10X); primary antibodies against caspase 9, caspase 3, cleaved caspase 3, Bcl-XL, Bax, and β-actin; and horseradish peroxidase (HRP)-conjugated secondary antibody from rabbit and mouse were obtained from Cell Signaling Technology (Danvers, MA, USA).

### 3.2. Polar Steroids from P. pectinifera

Compounds (25*S*)-5α-cholestane-3β,4β,6α,7α,8,15β,16β,26-octaol (**1**), (25*S*)-5α-cholestane-3β,6α,7α,8,15α,16β,26-heptaol (**2**), and asterosaponin P_1_ (**3**) were isolated from the starfish *Patiria* (=*Asterina*) *pectinifera* as described previously and were pure according to NMR, MS, TLC, and HPLC data [37]. Polar steroids were dissolved in DMSO as stock solutions (20 mM) and then diluted in culture media.

### 3.3. Cell Lines and Culture

Human colorectal carcinoma DLD-1 cells (ATCC® CCL-221™), HCT 116 (ATCC® CCL-247™), and HT-29 (ATCC® HTB-38™), were obtained from the American Type Culture Collection (Manassas, VA, USA).

DLD-1 cells were cultured in an RPMI-1640 medium, and HCT 116 and HT-29 were cultured in McCoy’s 5A medium supplemented with FBS (final concentration 10%) and penicillin-streptomycin solution (final concentration 1%). The cell cultures were maintained at 37 °C in a humidified atmosphere containing 5% CO_2_. Every 3–4 days, the cells were rinsed with PBS and treated with 0.25% trypsin/0.05 M EDTA for 1–3 min. Then 10%–20% of the harvested cells were transferred to a new flask containing fresh complete culture media.

### 3.4. MTS Assay

The cytotoxic activity of polar steroids from *P. pectinifera* was determined as described earlier with slight modifications [38]. Briefly, colorectal carcinoma DLD-1, HCT 116, and HT-29 cells (1.0 × 10^4^/200 µL) were seeded in a 96-well plate and incubated for 24 h at 37 °C in a CO_2_ incubator. The cells were treated with either PBS (control) or polar steroids from *P. pectinifera*
**1**–**3** (5, 10, 50, and 150 µM) for 24 h. Subsequently, the cells were incubated with 15 µL MTS reagent for 3 h, and the absorbance of each well was measured at 490/630 nm using a Power Wave XS microplate reader (BioTek, Winooski, VT, USA).

### 3.5. Soft Agar Colony Formation Assay

The effect of the investigated compounds on the colony formation of cancer cells was conducted as previously reported with modifications [38]. Briefly, DLD-1, HCT 116, and HT-29 cells (2.4 × 10^4^/mL) were treated with compounds (10, 20, and 40 µM) and applied in the 0.3% BME agar containing 10% FBS, 2 mM L-glutamine, and 25 µg/mL gentamicin. The cultures were maintained at 37 °C in a 5% CO_2_ incubator for 14 days, and the number and size of the colonies were determined using a Motic microscope AE 20 (XiangAn, Xiamen, China) and ImageJ software bundled with 64-bit Java 1.8.0_112 (NIH, Bethesda, MD, USA).

### 3.6. X-ray Exposure

To expose the cells to X-ray radiation, the XPERT 80 X-ray system (KUB Technologies, Inc, Milford, CT, USA) was used. The DRK-1 X-ray radiation clinical dosimeter (Axelbant LLK, Moscow, Russia) was used to measure the absorbed dose of radiation.

### 3.7. Cells Irradiation Assay

In brief, to determine the sensitivity of the DLD-1, HCT 116, and HT-29 cells to radiation, cells (5.0 × 10^5^) were seeded in 60-mm dishes and incubated for 24 h. The cells were exposed to X-ray at a dose rate from 2 to 10 Gy. Then, the cells recovered at 37 °C in a 5% CO_2_ incubator for 3 h. The cells were harvested with 0.25% trypsin/0.05 M EDTA solution and subjected to the MTS assay as described above. 

To determine the radiosensitizing activity of polar steroids **1**–**3,** DLD-1, HCT 116, and HT-29 cell (4.0 × 10^5^) were seeded in 60-mm dishes and incubated for 24 h. Then, the cells were treated with either PBS (control) or compounds **1**–**3** (1, 2, and 4 μM) for an additional 48 h. Cells were exposed to 2 Gy of X-ray and incubated for 3 h. Then, the cells were harvested and used for the soft agar colony formation assay, as described above, as well as western blotting and DNA comet assays, as described below.

### 3.8. Western Blotting Assay

The preparation of the cells lysate and western blotting assay were described in the work [38]. Briefly, the protein content was determined by the DC protein assay (Bio-Rad, Hercules, CA, USA). Lysate protein (30 µg) was subjected to 12% SDS-PAGE and electrophoretically transferred to a polyvinylidene difluoride membranes (PVDF) (Millipore, Burlington, MA, USA). The membranes were blocked with 5% non-fat milk (Bio-Rad) for 1 h and then incubated with the respective specific primary antibody at 4 °C overnight. Protein bands were visualized using an enhanced chemiluminescence reagent (ECL) (Bio-Rad, Hercules, CA, USA) after hybridization with an HRP-conjugated secondary antibody. Band density was quantified using the Image Lab™ 4.1 software (Bio-Rad).

### 3.9. DNA Comet Assay

The irradiated HT-29 cells (1 × 10^3^) were mixed with 0.5 mL of 1% low-melting-point agarose and applied to agarose-precoated slides. To detect single- and double-stranded DNA breaks, cell lysis and agarose gel electrophoresis were performed under alkaline conditions at 15 V for 30 min at 4 °C, as described previously [38]. The agarose gel was stained with ethidium bromide (25 µg/mL), and DNA comets were visualized using a ZOE™ Fluorescent Cell Imager fluorescent microscope (Bio-Rad). Received images were processed using the CaspLab/Comet assay software version 1.2.3beta2 (http://casplab). The tail moment was used as a parameter of the comets, which is defined as the percentage of DNA in the tail multiplied by the length between the center of the head and tail [39]. The tail moment of fifty DNA comets from each group was calculated.

### 3.10. Statistical Analysis

All assays were performed in at least three independent experiments. Results are expressed as the mean ± standard deviation (SD). The Student’s *t*-test was used to evaluate the data with the following significance levels: **p* < 0.05, ***p* < 0.01, ****p* < 0.001.

## 4. Conclusions

The inhibitory effect of polar steroids from starfish *P. pectinifera* on colony formation in colorectal carcinoma cells and the sensitivity of tested cells to radiation were investigated. The radiosensitizing activity of steroidal monoside asterosaponin P_1_ from *P. pectinifera*, which reduced the number and size of the colonies of human colorectal carcinoma cells, was discovered for the first time. Asterosaponin P_1_ was demonstrated to significantly enhance the radiation-induced apoptosis of HT-29 cells by the regulation of anti- and pro-apoptotic protein expression followed by the activation of the initiator and effector caspases and DNA degradation.

Based on the obtained data, we may assume that polar steroids from starfishes can be promising candidates for chemoradiation therapy, but more detailed investigations on the molecular mechanism of their action, as well as animal studies, are needed.

## Figures and Tables

**Figure 1 molecules-24-03154-f001:**
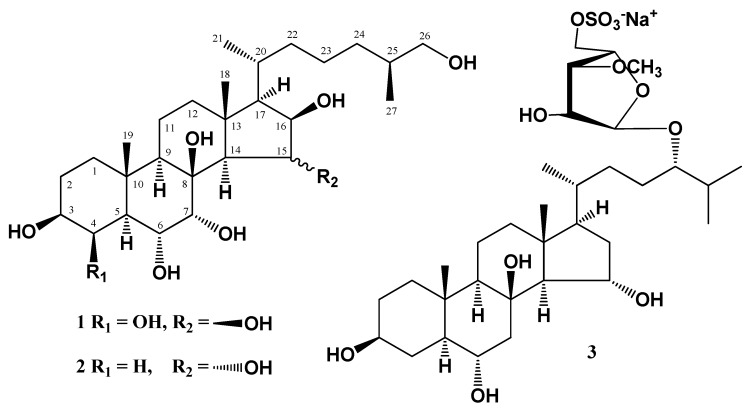
The structure of polar steroids isolated from starfish *P. pectinifera.*

**Figure 2 molecules-24-03154-f002:**
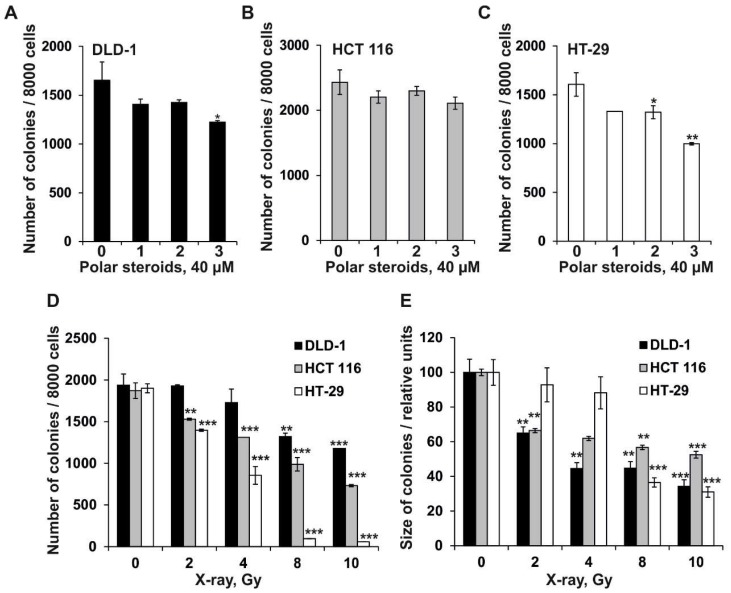
The effect of polar steroids from *P. pectinifera* (**1**–**3**) and X-ray radiation on colony formation in human colorectal carcinoma cells. DLD-1 (**A**), HCT 116 (**B**), and HT-29 (**C**) cells (2.4 × 10^4^) with or without polar steroid **1**–**3** (40 µM) or (**D**,**E**) X-ray radiation (2–10 Gy) treatment were subcultured onto 0.3% Basal Medium Eagle (BME) agar containing 10% FBS, 2 mM L-glutamine, and 25 µg/mL gentamicin. After 14 days of incubation, the number (**A**–**D**) and size (**E**) of the colonies were evaluated under a microscope with the aid of the ImageJ software program. All experiments were repeated at least three times in each group (n = 9 for control or compounds treated cells or X-ray exposed cells, n—quantity of photos). Results are expressed as the mean ± standard deviation (SD). The asterisk (*) indicates a significant decrease in the number or size of the colonies of cancer cells treated by polar steroids or X-ray compared to PBS-treated cells (**p* < 0.05, ***p* < 0.01, ****p* < 0.001).

**Figure 3 molecules-24-03154-f003:**
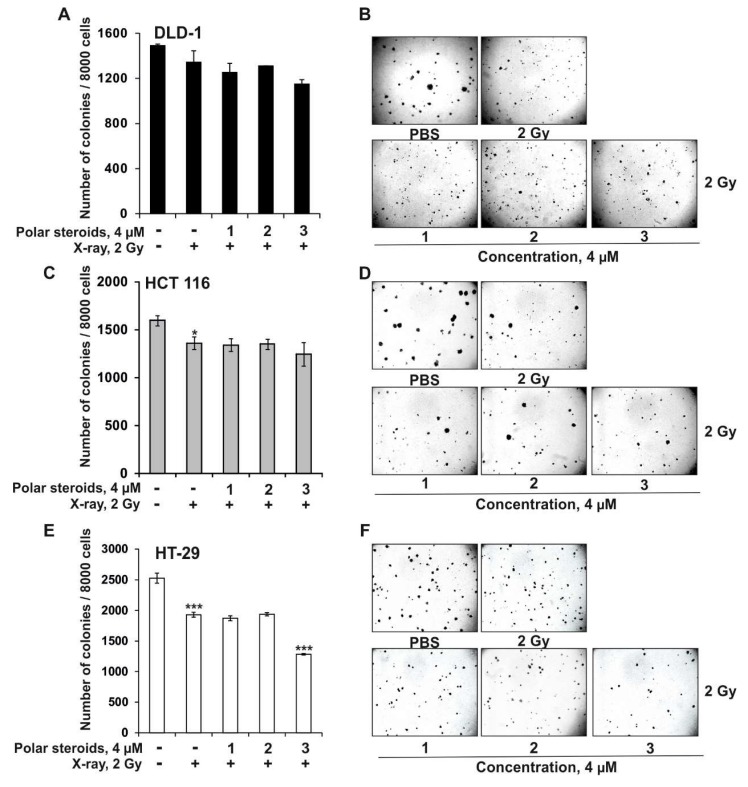
The radiosensitizing effects of polar steroids from *P. pectinifera* (**1**–**3**) on colony formation in human colorectal carcinoma cells. DLD-1 (**A**,**B**), HCT 116 (**C**,**D**), and HT-29 (**E**,**F**) cells (2.4 × 10^4^) were either treated or not treated with a combination of polar steroids **1**–**3** (1, 2, 4 µM) and X-ray radiation (2 Gy) and subcultured onto 0.3% Basal Medium Eagle (BME) agar containing 10% FBS, 2 mM L-glutamine, and 25 µg/mL gentamicin. After 14 days of incubation, the number (**A**,**C**,**E**) and size (**B**,**D**,**F**) of the colonies was evaluated under a microscope with the aid of the ImageJ software program. All experiments were repeated at least three times in each group (n = 9 for control or compounds treated cells or X-ray exposed cells, n—quantity of photos). The magnification of representative photos is 10×. Results are expressed as the mean ± standard deviation (SD). The asterisk (*) indicates a significant decrease in the number or size of colonies of cancer cells treated with X-ray compared to PBS-treated cells or polar steroids in combination with X-ray compared to irradiated cells (**p* < 0.05, ****p* < 0.001).

**Figure 4 molecules-24-03154-f004:**
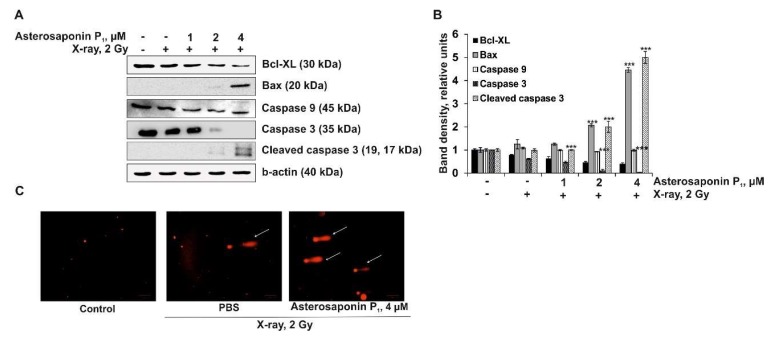
The effect of combinatorial treatment by asterosaponin P_1_ from *P. pectinifera* with radiation on apoptosis induction in HT-29 cells. (**A**) Regulation of the anti-apoptotic (Bcl-XL) and pro-apoptotic (Bax) proteins expression as well as the initiator (caspase 9) and effector (caspase 3, cleaved caspase 3) caspases, and b-actin by X-ray or with a combination of asterosaponin P_1_ and X-ray in HT-29 cells. (**B**) Relative bands density was measured using Image Lab™ Software 4.1. Quantitative results are presented as the mean values from three independent experiments. Significant differences were evaluated using the Student’s *t*-test. The asterisks (****p* < 0.001) indicate a significant alteration of protein expression in cells treated by X-ray compared to PBS-treated cells, or asterosaponin P_1_ in combination with X-ray compared to irradiated cells. (**C**) HT-29 cells were either treated or not treated by X-ray (2 Gy) or with a combination of asterosaponin P1 (4 µM) and radiation (2 Gy). DNA comets were stained with ethidium bromide and visualized using a fluorescent microscope ZOE™ Fluorescent Cell Imager. Representative images (Scale bar = 100 µm).

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
