# Peer review of "Effects of Polar Steroids from the Starfish Patiria (=Asterina) pectinifera in Combination with X-Ray Radiation on Colony Formation and Apoptosis Induction of Human Colorectal Carcinoma Cells"

_molecules, 2019, doi:10.3390/molecules24173154_

Round 1
Reviewer 1 Report
Dear authors,
The study is intersting and valuable.
In order to be published you need to work and improve the following according to the lines :
95-sentence not clear
100,101-English
104-be more models
109-english
115-colonies instead of cells
147-pucture are described in the legend but there are no pictures shown n figure 2
149-need to write compared to cells
158-to and not with
194-with and not by
202-203-English not clear
212-add cell after sensitive
216-217- this sentence tell nothing-can be omitted
220-English
244-what do you mean by expression balance between?
249-250- sound like the experiment have been done already. but not.
258-need to be modest and careful and write: we suggest that...
302-309- the description of the metho need to be in materials and methods. Too much description for the result part.
377-it is enough once to describe BioRad
398-English (need by)
Author Response
Dear Reviewer!
Thank you for careful review of our manuscript ID: molecules-566469 “Effects of polar steroids from the starfish Patiria (=Asterina) pectinifera in combination with X-ray radiation on colony formation and apoptosis induction of human colorectal carcinoma cells”. We are very grateful for your censorious remarks and useful comments. All comments of reviewer were accepted and manuscript was corrected according these comments. Notably:
95-sentence not clear
Answer. We agree with reviewer’s comment. The sentence was corrected as “Colony formation is an ability of a single cancer cell to form colony via clonal expansion and is thought to be one of the chemotherapeutic hallmarks of carcinogenesis” and reference [Blumenthal, R. D.; Goldenberg, D. M. Methods and goals for the use of in vitro and in vivo chemosensitivity testing. Mol. Biotechnol. 2007, 35(2), 185-197. doi: https://doi.org/10.1007/BF02686104] was added.
100,101-English
Answer. We agree with reviewer’s comment. The sentence was corrected.
104-be more models
Answer. We agree with reviewer’s comment. The sentence was corrected as “Among polar steroids investigated in this study, steroidal monoside asterosaponin P1 had the highest inhibitory activity against colony formation in HT-29 cells”.
109-english
Answer. We agree with reviewer’s comment. The sentence was corrected.
115-colonies instead of cells
Answer. We agree with reviewer’s comment. Corrected as suggested by reviewer.
147-pucture are described in the legend but there are no pictures shown n figure 2
Answer. We agree with reviewer’s comment. To calculate the number and size of colonies we made at least 9 photos from each experimental group. The figure legend was corrected. The sentence “All experiments were repeated at least three times in each group (n = 9 for control or compounds treated cells or X-ray exposed cells, n – quantity of photos)” was added.
149-need to write compared to cells
Answer. We agree with reviewer’s comment. Corrected as suggested by reviewer.
158-to and not with
Answer. We agree with reviewer’s comment. Corrected as suggested by reviewer.
194-with and not by
Answer. We agree with reviewer’s comment. Corrected as suggested by reviewer.
202-203-English not clear
Answer. We agree with reviewer’s comment. The sentence was corrected.
212-add cell after sensitive
Answer. We agree with reviewer’s comment. Corrected as suggested by reviewer.
216-217- this sentence tell nothing-can be omitted
Answer. We agree with reviewer’s comment. The sentence was deleted from the text.
220-English
Answer. We agree with reviewer’s comment. The sentence was corrected.
244-what do you mean by expression balance between?
Dear reviewer, thank you for the valuable remark. This is mistaken expression. The sentence was corrected as “Steroidal glycosides from the starfish Anthenea aspera exhibited pro-apoptotic activity via the regulation of expression level of pro-survival protein Bcl-XL and pro-apoptotic proteins Bax and Bak, followed by the activation of initiator and effector caspases in colon cancer cells”.
249-250- sound like the experiment have been done already. but not.
Answer. We agree with reviewer’s comment. The sentence was corrected as “Since polar steroids from starfishes induced apoptosis in different types of cancer cells, we checked the hypothesis whether asterosaponin P1 from P. pectinifera enhanced the radiation-induced apoptosis in colorectal carcinoma HT-29 cells by the activation of major players in mitochondrial apoptosis (Bax, Bcl-XL, caspase 9, and caspase 3)”.
258-need to be modest and careful and write: we suggest that...
Answer. We agree with reviewer’s comment. We apologize for the unambiguous conclusions. The sentence corrected as suggested by reviewer.
302-309- the description of the metho need to be in materials and methods. Too much description for the result part.
Answer. We agree with reviewer’s comment. The methodological part was deleted from the part of “Results and discussion”.
377-it is enough once to describe BioRad
Answer. Agree with reviewer’s comment. Corrected as suggested by reviewer.
398-English (need by)
Answer. Agree with reviewer’s comment. Corrected as suggested by reviewer.
Reviewer 2 Report
In this study, the authors examined the anti-cancer effects of polar steroids in colorectal cancer cells in vitro. Radiation therapy for colorectal cancer is considered to be limited, and so the strategies of combination treatment are expected. While basic research for that combination therapy is relevant, there are several issues to be solved in this research.
In this study, the authors focused on the mitochondrial apoptotic pathway induced by asterosaponin P1 in HT-29 cells. Is there any basis for denying the involvement of the caspase-8 dependency pathway in their condition? Caspase-8 is well known to have a central role for initiating apoptosis through the death receptors activated by extrinsic factors including radiation. It is also reported that caspase-8 is detected in HT-29 cells employed in this study.
In figure 2D, irradiation of 2 and 4 Gy did not significantly reduce the cell number of DLD-1. However, in figure 3A, the DLD-1 cell number irradiated with 2 Gy alone was significantly low compared to control. The authors should clearly describe the contradictions of these findings. The results in other cell lines are also considered to be not stable.
In figure 4C, the authors examined the effect of X-ray/asterosaponin P1 by using comet assay. In evaluation for comet assay, comet area, comet length, tail length, etc. are used as significant parameters. It needs to be clarified about the evaluation method for comet assay.
The effects on the cells were examined for 14 days in figure 2 and 3. When were polar steroids additions and x-rays given? The authors should describe the methods more carefully for the reader's understanding. Similarly, how long did the experiment in Fig 4 take place?
The Results and Discussion sections should be independent.
Author Response
Dear Reviewer!
Thank you for careful review of our manuscript ID: molecules-566469 “Effects of polar steroids from the starfish Patiria (=Asterina) pectinifera in combination with X-ray radiation on colony formation and apoptosis induction of human colorectal carcinoma cells”. We are very grateful for your censorious remarks and useful comments. All comments of reviewer were accepted and manuscript was corrected according these comments. Notably:
In this study, the authors focused on the mitochondrial apoptotic pathway induced by asterosaponin P1 in HT-29 cells. Is there any basis for denying the involvement of the caspase-8 dependency pathway in their condition? Caspase-8 is well known to have a central role for initiating apoptosis through the death receptors activated by extrinsic factors including radiation. It is also reported that caspase-8 is detected in HT-29 cells employed in this study.
Answer. Dear reviewer, thank you very much for your valuable comment.
In the present study we focused on the investigation of combinatorial effect of polar steroids and X-ray on induction of apoptosis via intrinsic (mitochondrial) pathway, because previously it was found out that polar steroids from starfishes induced apoptosis by the regulation of proteins expression of mitochondrial pathway [Zhou, J.; Cheng, G.; Tang, H.F.; Zhang, X. Novaeguinoside II inhibits cell proliferation and induces apoptosis of human brain glioblastoma U87MG cells through the mitochondrial pathway. Brain Res. 2011, 1372, 22-28. doi: 10.1016/j.brainres.2010.11.093.], [Malyarenko, T.V.; Malyarenko, O.S.; Kicha, A.A.; Ivanchina, N.V.; Kalinovsky, A.I.; Dmitrenok, P.S.; Ermakova, S.P.; Stonik, V.A. In Vitro Anticancer and proapoptotic activities of steroidal glycosides from the starfish Anthenea aspera. Mar. Drugs 2018, 16(11), E420. doi: 10.3390/md16110420]. In this work we did not investigate the effect of polar steroids on induction of apoptosis through extrinsic (death receptor) pathway. The deeper elucidation of the molecular mechanism of radiosensitizing activity of polar steroids from starfishes including extrinsic pathway of apoptosis is the aim of our future work.
In figure 2D, irradiation of 2 and 4 Gy did not significantly reduce the cell number of DLD-1. However, in figure 3A, the DLD-1 cell number irradiated with 2 Gy alone was significantly low compared to control. The authors should clearly describe the contradictions of these findings. The results in other cell lines are also considered to be not stable.
Answer. We carefully checked all calculations regarding to colony formation assay. The data represented in Figure 2 (A–E) is correct. The graphs of Figure 3 (A and E) have some errors. We are very sorry for these errors. In particular, in Figure 3A standard deviation (SD) was recalculated and corrected. In this case the number of colonies of DLD-1 cells reduced by 8%±7 after X-ray exposure of 2 Gy. In figure 2D, X-ray irradiation (2 Gy) caused the decreasing of the colonies number of DLD-1 cells by 2%±1. Such minor differences in the values can be justified by the experimental error. In Figure 3E there was mistake in calculations of number of colonies of HT-29 cells irradiated by 2 Gy. Figure 3 (A and E) was corrected.
In figure 4C, the authors examined the effect of X-ray/asterosaponin P1 by using comet assay. In evaluation for comet assay, comet area, comet length, tail length, etc. are used as significant parameters. It needs to be clarified about the evaluation method for comet assay.
Answer. We agree with reviewer’s comment. The quantitative analysis for comet assay was conducted using CaspLab/Comet Assay Software version 1.2.3beta2. Tail moment was used as a parameter of comets which is defined as the percentage of DNA in the tail multiplied by the length between the center of the head and tail [Olive, P. L.; Banath, J. P.; Durand, R. E. Heterogeneity in radiation induced DNA damage and repair in tumor and normal cells using the “Comet” assay. Radiat. Res. 1990, 122, 86–94].
The effects on the cells were examined for 14 days in figure 2 and 3. When were polar steroids additions and x-rays given? The authors should describe the methods more carefully for the reader's understanding.
Answer. At first the cells were treated by polar steroids for 48 h and then cells were irradiated by X-ray (2 Gy). After 3 h, cells were subjected to the soft agar assay. To form colonies, treated by compounds and irradiated cancer cells were incubated for additionally 14 days. These procedures described in details in the Section “3.7. Cells irradiation assay” of Material and Methods.
Similarly, how long did the experiment in Fig 4 take place?
Answer. HT-29 cells were seeded in 60-mm dishes and incubated for 24 h. Then, cells were treated with either PBS (control) or compound 3 (1, 2, and 4 μM) for an additional 48 h. Cells were exposed to 2 Gy of X-ray and incubated for 3 h. Then, the cells were harvested and used for comet assays. So totally, experiment in Fig 4C took place for 75 h.
The Results and Discussion sections should be independent.
Answer. We agree with reviewer’s comment that independent sections “Results” and “Discussion” may be better for reader's understanding, however according to the Instruction for Authors of Molecules the section “Discussion” may be combined with “Results”. We combined these sections because there was not much information about radiosensitizing activity of polar steroids from starfishes in the literature.
Reviewer 3 Report
The manuscript entitled "Effects of polar steroids from the starfish Patiria (=Asterina) pectinifera in combination with X-ray radiation on colony formation and apoptosis induction of human colorectal carcinoma cells" by Malyarenko et al focused the attention on the anticancer and radiosensitizing role of polar steroids form P. pectinifera in colorectal cancer cell model.
The manuscript was well written and suitable for publication after minor revision:
- In the introduction section page 1 lines 36-39 "Radiation therapy using high-energy rays (such as X-ray) is the method most often used to destroy rectal cancer cells. In turn, combinatorial treatment by chemo- and radiation therapy, called chemoradiation therapy, is the most effective curative modality for people with colorectal cancer [2]." The Authors should re-phrase taking into account the major role of surgery in the managment of colorectal cancer patients.
- Page 3 line 95 "Colony formation is one of the chemotherapeutic hallmarks of cell transformation" could the Authors add a references for this statement?
Author Response
Dear Reviewer!
Thank you for careful review of our manuscript ID: molecules-566469 “Effects of polar steroids from the starfish Patiria (=Asterina) pectinifera in combination with X-ray radiation on colony formation and apoptosis induction of human colorectal carcinoma cells”. We are very grateful for your censorious remarks and useful comments. All comments of reviewer were accepted and manuscript was corrected according these comments. Notably:
- In the introduction section page 1 lines 36-39 "Radiation therapy using high-energy rays (such as X-ray) is the method most often used to destroy rectal cancer cells. In turn, combinatorial treatment by chemo- and radiation therapy, called chemoradiation therapy, is the most effective curative modality for people with colorectal cancer [2]." The Authors should re-phrase taking into account the major role of surgery in the managment of colorectal cancer patients.
Answer. We agree with reviewer’s comment. The sentence was rephrased as “In turn, surgery, chemotherapy, immunotherapy, and combinatorial treatment by chemo- and radiation therapy, called chemoradiation therapy are the most effective curative modality for people with colorectal cancer [2].”
- Page 3 line 95 "Colony formation is one of the chemotherapeutic hallmarks of cell transformation" could the Authors add a references for this statement?
Answer. We agree with reviewer’s comment. The reference [Blumenthal, R. D.; Goldenberg, D. M. Methods and goals for the use of in vitro and in vivo chemosensitivity testing. Mol. Biotechnol. 2007, 35(2), 185-197] was added.
Round 2
Reviewer 2 Report
In the response, the authors described that this compound exerts cytotoxicity against different cancer cell lines without selectivity. This finding is considered to be very important to know the mechanism of toxicity of this compound. The authors should describe the existence of Nur77-independent toxicity in several types of cells. Furthermore, it should be noted that the Nur77-dependent toxicity of this compound can only be confirmed with HeLa.
Author Response
In the text of manuscript we wrote that “None of the tested compounds inhibited the viability of HCT 116, DLD-1, and HT-29 cells by 50% at concentrations up to 150 µM” (Lines 77 - 78). It means that investigated compounds do not exert potent cytotoxic activity against colorectal carcinoma cells at tested concentrations. It is not completely clear why reviewer asked to investigate whether polar steroids possessed Nur77- independent or -dependent toxicity?
In our study we aimed to study the anticancer and radiosensitizing activities of polar steroids from P. pectinifera in a colorectal carcinoma cell model of colony formation and apoptosis induction.
It is known that Nur77 induced by certain apoptotic stimuli, and on appropriate modification (probably by phosphorylation and dephosphorylation), migrates from the nucleus to the cytoplasm as Nur77/RXR heterodimer, where it targets mitochondria by binding to Bcl-2. Interaction with Bcl-2 induces a Bcl-2 conformational change, triggering cytochrome c release and apoptosis [Zhang, X. Targeting Nur77 translocation. Expert Opin. Ther. Targets. 2006, 11(1), 69–79. doi:10.1517/14728222.11.1.69].
We showed that asterosaponin P1 was able to sensitize colorectal carcinoma cells HT-29 to X-ray radiation thorough the regulation of expression of several proteins which are members of mitochondrial pathway. Surely, it is interesting to check the idea whether polar steroids can influence Nur77 translocation and induce apoptosis, but this is different project that may be implemented in the future work.